# Evaluation of Intracranial Hypertension in Traumatic Brain Injury Patient: A Noninvasive Approach Based on Cranial Computed Tomography Features

**DOI:** 10.3390/jcm10112524

**Published:** 2021-06-07

**Authors:** Yingchi Shan, Yihua Li, Xuxu Xu, Junfeng Feng, Xiang Wu, Guoyi Gao

**Affiliations:** 1Department of Neurosurgery, Shanghai General Hospital, Shanghai Jiao Tong University School of Medicine, Shanghai 201600, China; 18301965008@163.com (Y.S.); liyihuamail2020@163.com (Y.L.); drwuxiang@163.com (X.W.); 2Department of Neurosurgery, Renji Hospital, Shanghai Jiao Tong University School of Medicine, Shanghai 201600, China; 18326936939@163.com (X.X.); fengjfmail@163.com (J.F.)

**Keywords:** traumatic brain injury, computed tomography, intracranial pressure, noninvasive intracranial pressure evaluation

## Abstract

Background: Our purpose was to establish a noninvasive quantitative method for assessing intracranial pressure (ICP) levels in patients with traumatic brain injury (TBI) through investigating the Hounsfield unit (HU) features of computed tomography (CT) images. Methods: In this retrospective study, 47 patients with a closed TBI were recruited. Hounsfield unit features from the last cranial CT and the initial ICP value were collected. Three models were established to predict intracranial hypertension with Hounsfield unit (HU model), midline shift (MLS model), and clinical expertise (CE model) features. Results: The HU model had the highest ability to predict intracranial hypertension. In 34 patients with unilateral injury, the HU model displayed the highest performance. In three classifications of intracranial hypertension (ICP ≤ 22, 23–29, and ≥30 mmHg), the HU model achieved the highest F1 score. Conclusions: This radiological feature-based noninvasive quantitative approach showed better performance compared with conventional methods, such as the degree of midline shift and clinical expertise. The results show its potential in clinical practice and further research.

## 1. Introduction

Traumatic brain injury (TBI) is a major public health problem around the world. It is estimated that 69 million individuals worldwide suffer a TBI each year, leading to death or life-long disability [1]. Clinically, the intracranial pressure (ICP) level is the major intuitive indicator in neural functional evaluation, and the monitoring of ICP has a great significance both in identifying surgical indication and in critical care management [2]. Invasive techniques, for example, the placement of an ICP probe into the cranium, are the main methods of ICP monitoring. Despite being precise and accurate, this invasive treatment not only requires costly equipment but also leads to complications including intracranial hemorrhage, intracranial infection, and brain parenchymal injury, which in rare cases can cause severe danger to the patients [3,4]. It is reasonable that noninvasive methods, including acoustic elasticity, optic nerve sheath diameter, and transcranial Doppler, exhibit potential in ICP evaluation with fewer complications [5]. Meanwhile, as a noninvasive examination for craniocerebral trauma, computed tomography (CT) scans play a vital role in the disease diagnosis, severity evaluation, and outcome prediction [6,7]. Until now, the information acquired from CT images has been restricted empirically. The implementation of CT images to assess ICP still relies mostly on the experience of clinicians [8], and a quantitative and accurate method is lacking [9]. The purpose of this study was to noninvasively evaluate the ICP of patients in an early phase before surgery using the Hounsfield unit (HU) method. This method was used to establish a model by extracting the HU value and its related features, which can reflect the symmetry, uniformity, and local intensity distribution changes of CT images, so as to evaluate intracranial pressure. In addition, two traditional methods, using midline shift and clinical expertise, were included in this study, and the ability of these three methods to predict intracranial hypertension was compared.

## 2. Materials and Methods

### 2.1. Study Design and Setting

In this retrospective study, the clinical data of patients with TBI admitted to the Department of Neurosurgery, Shanghai General Hospital from January 2018 to December 2019 were collected and analyzed. Inclusion criteria were as follows: (1) patients with an acute closed TBI; (2) received invasive ICP monitoring according to the guidelines for the management of severe traumatic brain injury; (3) received emergency cranial CT scan within 60 min of ICP monitoring; (4) 18–65 years old. Exclusion criteria were as follows: patients with a history of TBI, cerebral infarction, brain tumors, or other neurological diseases or cranial surgical interventions that might result in an abnormal anatomical structure or CT density. The study protocol conformed to the ethical guidelines of the Declaration of Helsinki, and this study was approved by the Ethics Committee of Shanghai General Hospital, Shanghai Jiao Tong University School of Medicine (RA-2019-180). Participants’ right to know was fully guaranteed and indicated in the ethical approval document.

### 2.2. Data Sources and Measurements

In addition to demographic and baseline injury characteristics, we collected and analyzed the last cranial CT before ICP monitoring, as well as the initial ICP value immediately after ICP sensor insertion, before performing a craniotomy for hematoma removal. Due to transportation and the anesthetic preparation of patients, the duration between the last cranial CT and ICP monitoring was about 30–60 min. In this study, the ICP sensor was inserted into the ventricle, and the location of the sensor was verified by cranial CT after surgery. The initial ICP was recorded during operation immediately after the insertion of the ICP sensor (Integra, Plainsboro, NJ, USA). The Hounsfield unit (HU) values were acquired from the Digital Imaging and Communications in Medicine (DICOM) file of the last cranial CT before ICP monitoring using a 64-slice spiral CT machine (General Electric Medical Systems, Fairfield, CT, USA). As per the routine protocol of a CT scan, the CT slices were parallel to the orbitomeatal plane from the foramen magnum to the vertex. The scanning slice thickness was 5 mm. The slice with the body of the lateral ventricle (50 mm above the basal plane) was selected for the examination of Hounsfield unit (HU) and midline shift. Brain tissue was divided into lines perpendicular to the cranial midline after removing the interference of the skull, and the HU value on the line across the anterior pole of the lateral ventricle was collected. HU-related features extracted included (1) the mean of HU on each side of the cranial midline and their ratio, (2) the standard deviation of HU on each side and their ratio, (3) the Shannon entropy of HU on each side and their ratio, (4) and total Shannon entropy, which included both sides in the calculation. All ratios were defined as the division of the larger value and smaller value to ensure values ≥1. Midline shift was measured in millimeters, as the perpendicular distance between the maximum displacement of the septum pellucidum and cranial midline (median ridge of the fronto-parietal bone). In addition, three experienced neurosurgeons (unaware of the initial ICP) were asked to read the CT slide and evaluate whether there was intracranial hypertension (ICP > 22 mmHg), and the judgement was made after discussion and agreement.

Then, three models were established on the basis of HU features (HU model), the midline shift (MLS model), and clinical expertise (CE model). The establishment and analysis process of these three models is shown in Figure 1. In the HU model, the HU-related features introduced previously were included in multivariate logistic regression to predict intracranial hypertension (ICP > 22 mmHg). Considering the relatively small sample size, L2 regularization (ridge regression) was used to regularize the logistic regression model of HU-related features to reduce overfitting [10]. Briefly, after scaling, the L2 norm of the weight vector was added to the cost function, and regularization parameter lambda was chosen to minimize the mean cross-validated error. The performance of the regularized version of the model was also evaluated. The MLS model was established on the basis of midline shift data, using univariate logistic regression. The CE model was established as an analysis of the correlation between clinical judgment and the actual ICP value.

In consideration of the heterogeneity of TBI, patients’ data with focal lesions restricted in the unilateral hemisphere were extracted. Likewise, HU, MLS, and CE models were fitted in the same way to discriminate intracranial hypertension (ICP > 22 mmHg) in this subgroup. Then, the performance of each model was compared.

To achieve a more precise prediction, in addition to the binary classification of intracranial hypertension, these three models were used to classify ICP into three groups (≤22, 23–29, and ≥30 mmHg), and their respective performance was compared.

### 2.3. Statistical Analysis

Statistical analysis was performed on the data using R statistical software (version 3.5.0). Continuous variables subject to normal distribution were expressed as the mean ±SD, continuous variables not subject to normal distribution were expressed as the median (M) and interquartile range (IQR), and categorical variables were expressed as the frequency and percentage. The logistic regression was modeled using the “glm” function of the “stats” package (version 3.6.1). The regularized logistic regression model was established using the “glmnet” function of the “glmnet” package (version 3.0). The ordered logistic regression was modeled using the “plor” function of the “MASS” package (version 7.3). The area under the ROC curve was calculated using the “roc” function of the “pROC” package. A *p*-value < 0.05 was considered statistically significant.

Accuracy, precision, recall, and F1 score were used to evaluate the performance of each model. The area under the receiver operator characteristic (ROC) curve was used in all three models to assess discrimination. Lastly, the performance of the three models was compared.

## 3. Results

### Comparative Results of the Three Models

From January 2018 to December 2019, 47 patients were enrolled in this study, among which 30 were male (63.83%) and 17 were female (36.17%). Participants were between 36 and 65 years old, and the median age was 50 (IQR: 46–59) years old. The causes of injuries included traffic accidents (24, 51.06%), falls from heights (8, 17.02%), and falling (15, 31.91%). The median Glasgow Coma Score (GCS) was 7 (IQR: 5–8) at the time of emergency admission; 26 patients (55.32%) had an abnormal light reflex in one or both pupils upon arrival. The average duration between the last cranial CT and ICP monitoring was (40.55 ± 9.25) min. In these cases, 34 patients with focal lesions were listed as having a unilateral injury, whereas 13 patients were listed as having a bilateral injury. Furthermore, 21 of the enrolled patients (61.76%) suffered from a unilateral subdural hemorrhage, 10 patients (29.41%) suffered from a unilateral contusion, three patients (8.82%) suffered from a unilateral epidural hemorrhage, seven patients (53.85%) suffered from a bilateral contusion, four patients (30.77%) suffered from a subdural hemorrhage with contrecoup contusion, and two patients (15.38%) suffered from diffuse brain swelling.

As shown in Table 1, the median initial ICP of all participants was 25 mmHg (IQR: 20–30). The median midline shift was 4.17 mm (IQR: 1.17–6.02). The mean HU in each hemisphere ranged from 25.24 to 52.27, with an average of 34.78 ± 4.76, and the median ratio of both sides was 1.10 (IQR: 1.04–1.24). The standard deviation of HU on each side ranged from 4.43 to 21.17, with an average of 10.51 ± 2.96, and the median ratio of both sides was 1.25 (IQR: 1.09–1.55). The Shannon entropy of HU on each side ranged from 1.28 to 2.34, with a median of 1.78 (IQR: 1.64–1.98), and the median ratio of both sides was 1.11 (IQR: 1.05–1.21). The total Shannon entropy for each patient had a median value of 1.11 (IQR: 1.05–2.21).

The performance of the HU, MLS, and CE models in predicting intracranial hypertension is shown in Table 2 and Figure 2. The HU model had the highest ability to predict intracranial hypertension, with a threshold value of 22 mmHg. Although its performance decreased after regularization, the HU model still showed reasonable discrimination power. Both the original and the regularized model showed good calibration (χ^2^ = 2.83, df = 4, *p* = 0.59 and χ^2^ = 7.25, df = 4, *p* = 0.12, respectively), the plot of which is shown in Figure 3. The MLS model had limited power to discriminate intracranial hypertension. The CE model could help evaluate high ICP compared with the MLS model, but it was still inferior to the HU model. Subsequently, we evaluated the performance of the HU, MLS, and CE models in 34 patients with unilateral TBI (Table 3). The performance of both the HU and the MLS models improved. The original and regularized HU models had a high ability to predict intracranial hypertension (Figure 4), and both showed good calibration (χ^2^ = 2.13, df = 4, *p* = 0.71 and χ^2^ = 6.09, df = 4, *p* = 0.19, respectively; Figure 5). However, the MLS model still achieved the lowest discrimination ability. The CE model showed similar discrimination power in the analysis of the total patient population.

In the three classifications of ICP (ICP ≤22, 23–29, and ≥30 mmHg), the CE model failed to predict intracranial hypertension from the CT slide according to the clinicians’ expertise. The performance of the HU and MLS models is shown in Table 4, where it can be seen that HU-related features performed better for all three classifications of ICP compared with the MLS model.

## 4. Discussion

This retrospective study enrolled closed TBI patients, who received a cranial CT scan within 60 min of ICP monitoring to ensure the timeliness of CT images. It is demonstrated that the intracranial hypertension of TBI patients could be assessed noninvasively and quantitatively on the basis of Hounsfield unit features from a CT scan of an injured brain. The results showed that the proposed ICP assessment approach achieved higher accuracy in discriminating intracranial hypertension, compared with other methods estimating intracranial pressure, such as midline shift and clinical expertise.

Currently, the research and the clinical usage of cranial CT images in TBI patients mainly focus on CT scores, including the Marshall and Rotterdam CT score [11,12]. TBI patients are comprehensively classified on the basis of the radiological findings on CT images. According to those classifications, the relationship between CT and clinical severity or prognosis is extensively analyzed [13,14]. Several studies have focused on analyzing the relationship between CT image characteristics and ICP level [15,16]. Depending on the CT image, these studies determined the relationship between the degree of brain parenchymal compression or ventricle volume and ICP, in an attempt to describe the possibility of noninvasive ICP measurement; approaches to measuring the ventricle volume using the width of the third ventricle, perimesencephalic cistern, or sylvian fissure have also been indicated [17]. Compared with the CT score and intracranial volume, the HU method presented herein could better describe CT images and alleviate the subjective effects in scaling or quantifying. Clinical prediction based on a comparison of HU values in different regions was also reported in previous studies. For example, Kanazawa and colleagues detected HU-related parameters (mean CT value, entropy, skewness, and kurtosis in regions of interest (ROIs)) in the early postictal state to predict cerebral vasospasm, delayed cerebral ischemia, and functional outcome in aneurysmal SAH [18]. Shen and colleagues analyzed the image texture (mean gray-level intensity, variance, and uniformity of the ROI) of head CT scans to predict early hematoma enlargement [19]. Inaba and colleagues evaluated the brain density adjacent to the lateral ventricles on brain CT images as a predictor of elevated ICP with a negative result [20]. In our study, the method of comparing HU values between patients was modified. We analyzed the HU value on the line across the anterior pole of the lateral ventricle on each side of the brain and compared the differences between the two sides as a function of their ratio. This method allows one to magnify the differences in CT images compared to processing the local HU values.

After traumatic brain injury, increased intracranial pressure is one of the most common consequences of pathophysiological changes in the brain [21]. Recent evidence has indicated that, in TBI patients, an ICP above 22 mmHg is associated with increased mortality in adult patients [22], and it is recommended by guidelines to be treated. Even moderate to severe TBI patients with an ICP above 30 mmHg may indicate the presence of a cerebral hernia, which needs quick decompression [23]. Evaluating the ICP level at an early phase and, therefore, establishing treatment plans are among the most significant issues in TBI patient management, which is commonly based on clinical judgment according to neurofunctional evaluation and CT image interpretation or simply by referring to the degree of midline shift.

During intracranial hypertension, the brain tissue is deformed by hematoma or edema, and the stress is transmitted to the opposite side, which causes a midline shift. Then, intracranial pressure can be estimated according to the degree of midline shift. HU values, in contrast to the midline shift, provide more information about the structural and density changes on both sides of the brain. For instance, hematomas may present as high-density lesions while edemas or infarctions may result in low-density lesions on CT images. Clinicians typically make radiological diagnoses depending on the changes in HU, but its role in evaluating ICP is rarely studied. For the reasons above, we utilized a model based on HU-related parameters as a novel predictor of intracranial hypertension and compared it with commonly used methods, i.e., midline shift and clinical expertise [24].

In this study, features of the HU value were extracted, including average and variability. The mean HU reflects the centralized trend on both sides. The standard deviation reflects the variability and dispersion [25]. Furthermore, the Shannon entropy, which measures the complexity of a dataset, offers a different way to evaluate the variability of HU values [26,27]. After TBI, various pathological changes lead to an increase in HU variability, and the mean HU focuses more on the overall condition, which may lead to local influences being neglected. On the one hand, deviation indices can remedy the weakening effect of the mean HU; on the other hand, the inclusion of more indicators makes the fitting of the regression model more adequate.

According to the results, both the original and the regularized HU models attained relatively high F1 scores and AUCs, thus implying that features acquired from HU can discriminate TBI patients with intracranial hypertension. The findings indicated the better performance of the HU model compared to those based on midline shift and clinical experience. Although routinely used in clinical work, in this study, the MLS and CE models showed limited ability to assess ICP level. The degree of midline shift may be associated with many other factors including age, lesion type, and other pathological conditions, which may lead to unsatisfactory performance [28]. The intellectual judgement of intracranial hypertension involves the complex processing of individual experience in interpreting radiological findings. The results from this study indicated that the predictive power of clinical experience lacks stable and objective performance.

The results of this study imply that the heterogenetic pathologies of TBI may affect the performance of evaluation methods, especially in terms of the HU model and MLS model. For instance, the midline shift may be attenuated in TBI patients with bilateral lesions, which may further undermine the accuracy of ICP evaluation, thus compromising the judgment ability of the midline shift model. After an analysis of patients with unilateral injury, it was found that the performance of both the midline shift and the HU-related models improved, while the HU-related model retained the highest discrimination ability. This result serves to remind us that TBI is a disease with high heterogeneity, and the target cohort for evaluation or treatment should be selected to optimize the effectiveness.

In addition to binary classification, this study tried to carry out a more precise evaluation. The HU model also showed potential in categorizing ICP into three groups. However, clinical experts failed to reach an agreement for most patients, and the MLS model still showed limitations in the classification, thus rendering it incapable of more precise prediction. It should be noted that the predictive ability of HU-related features also decreased compared with its performance in binary classification. Therefore, subsequent studies should increase the sample size, extract additional CT features, and implement various machine learning algorithms to further improve the performance of the model.

This study had some limitations. Firstly, the number of cases was relatively small and from a single center, which may have led to the overfitting of the model. Secondly, the study degree evaluated a relatively short series of patients. The hemispheres of the brain may not be symmetrical, and the congenital anatomy is often ignored, which may lead to intrinsic errors in acquiring data. Thirdly, this study only selected the layer featuring the body of the lateral ventricle and only analyzed the HU extracted from the line across the brain, which cannot reflect the features of other layers. The CT images of patients were also affected by other factors such as the patient’s age and physical condition, which could lower the applicability of the model. Fourthly, the HU method evaluated ICP values in a semiquantitative way, without providing a continuous value. Fifthly, in this study, people aged between 18 and 65 were selected in order to exclude any bias caused by immature brain tissue in minors, as well as by age-related brain changes in the elderly; thus, the results cannot be applied to the global population. Lastly, the monitoring of ICP is a continuous and dynamic process. In this study, the initial ICP was assessed by the HU method, which could not reflect the ICP level after progression or treatment. These flaws need to be addressed in subsequent research.

## 5. Conclusions

In summary, this study revealed that the level of ICP can be assessed noninvasively and quantitatively by extracting HU-related features on both sides of the midline of the injured brain in a semiquantitative way. Compared with commonly used ICP assessment methods (i.e., the degree of midline shift and clinical expertise), this novel method could distinguish intracranial hypertension, allowing it to provide early warnings in clinical practice. Subsequent investigations should further investigate the best application of this method.

## Figures and Tables

**Figure 1 jcm-10-02524-f001:**
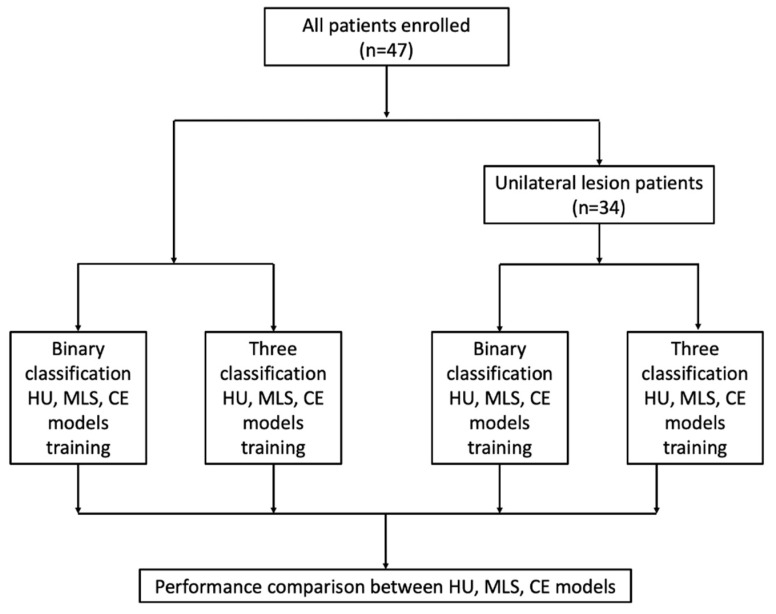
Model establishment and analysis process. HU, Hounsfield unit; MLS, midline shift; CE, clinical expertise.

**Figure 2 jcm-10-02524-f002:**
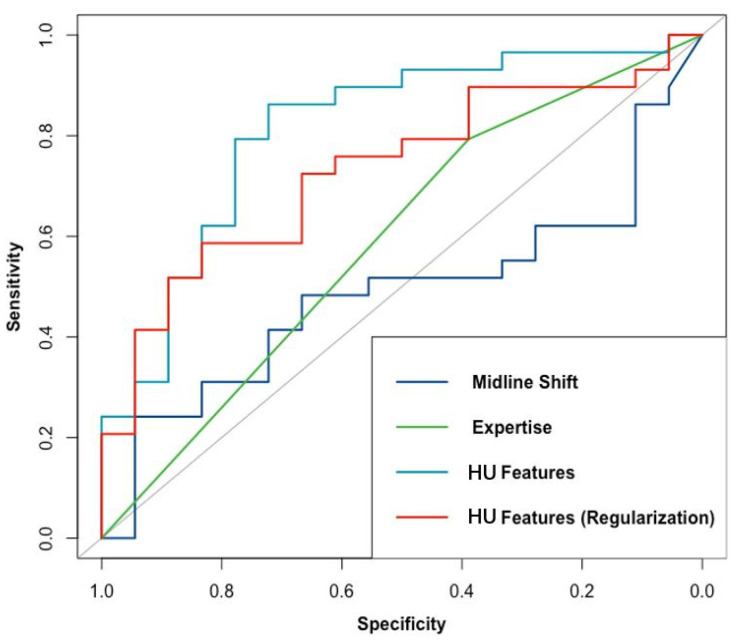
Performance of models based on HU features (original and regularized), midline shift, and clinical expertise in discriminating intracranial hypertension; the AUCs for these models were 0.81, 0.73, 0.49, and 0.59, respectively.

**Figure 3 jcm-10-02524-f003:**
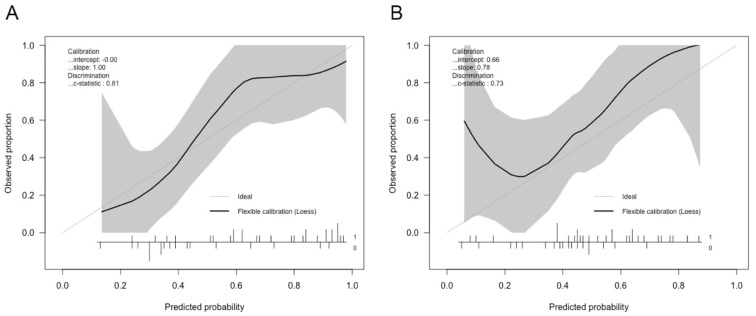
Calibration plot of HU feature-based models in discriminating intracranial hypertension: (**A**) calibration plot of logistic regression HU feature-based model; (**B**) calibration plot of regularized version of the model.

**Figure 4 jcm-10-02524-f004:**
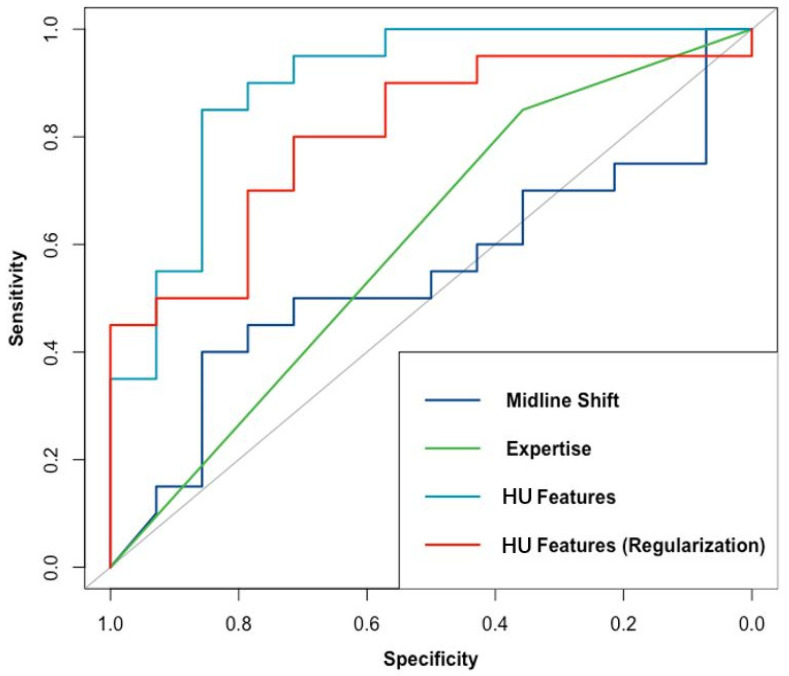
Performance of models based on HU features (original and regularized), midline shift, and clinical expertise in discriminating intracranial hypertension in unilateral TBI patients; the AUCs for these models were 0.90, 0.80, 0.54, and 0.60, respectively.

**Figure 5 jcm-10-02524-f005:**
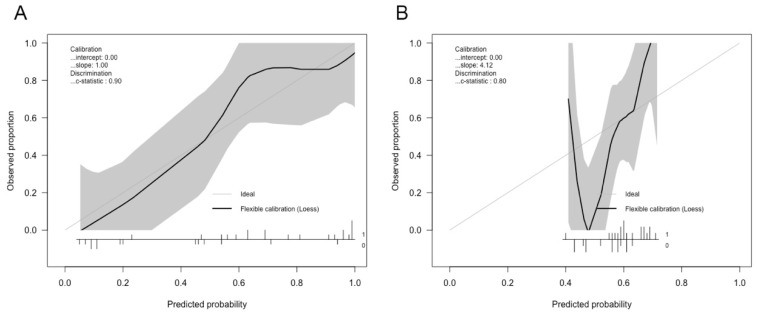
Calibration plot of HU feature-based models in discriminating intracranial hypertension in unilateral TBI patients: (**A**) calibration plot of logistic regression HU feature-based model; (**B**) calibration plot of regularized version of the model.

**Table 1 jcm-10-02524-t001:** Initial ICP- and CT-related characteristics of participants.

	All Patients with TBI (*n* = 47)	Patients with Unilateral TBI (*n* = 34)
Initial ICP (mmHg) ^a^	25 (20–30)	25 (19–30)
>22 mmHg	29 (62.07%)	20 (58.73%)
Midline shift (mm) ^a^	4.17 (1.17–6.02)	3.75 (1.15–6.06)
Mean HU		
Higher-value side ^b^	37.25 (4.87)	37.72 (5.20)
Lower-value side ^b^	32.31 (3.11)	32.77 (3.18)
Ratio ^a^	1.10 (1.04–1.24)	1.10 (1.03–1.25)
Standard deviation of HU		
Higher-value side ^b^	12.09 (3.06)	12.33 (3.43)
Lower-value side ^b^	8.93 (1.82)	8.77 (1.80)
Ratio ^a^	1.25 (1.09–1.55)	1.31 (1.12–1.65)
Shannon entropy of HU		
Total on both sides ^a^	1.94 (1.76–2.07)	1.94 (1.73–2.04)
Higher-value side ^a^	1.93 (1.78–2.05)	1.92 (1.75–2.04)
Lower-value side ^a^	1.69 (1.57–1.80)	1.66 (1.57–1.78)
Ratio ^a^	1.11 (1.05–1.21)	1.13 (1.05–1.19)

^a^ Variables presented as median (M) and interquartile range (IQR); ^b^ variables presented as mean and standard deviation.

**Table 2 jcm-10-02524-t002:** Performance of models based on HU features (original and regularized), midline shift, and clinical expertise in predicting intracranial hypertension.

	HU Features	HU Features (Regularized)	Midline Shift	Expertise
Accuracy	80.85%	65.96%	61.70%	63.83%
Precision	83.33%	84.21%	61.70%	67.65%
Recall	86.21%	55.17%	100.00%	79.31%
F1 Score	0.85	0.79	0.76	0.78
AUC (95% CI)	0.81 (0.68–0.94)	0.73 (0.58–0.88)	0.49 (0.32–0.66)	0.59 (0.45–0.73)

**Table 3 jcm-10-02524-t003:** Performance of models based on HU features (original and regularized), midline shift, and clinical expertise in unilateral TBI patients.

	HU Features	HU Features (Regularized)	Midline Shift	Expertise
Accuracy	85.29%	70.59%	58.82%	64.71%
Precision	85.71%	67.86%	58.82%	65.38%
Recall	90.00%	95.00%	100.00%	85.00%
F1 Score	0.88	0.79	0.74	0.74
AUC (95% CI)	0.90 (0.78–1.00)	0.80 (0.65–0.95)	0.54 (0.34–0.74)	0.60 (0.45–0.76)

**Table 4 jcm-10-02524-t004:** Performance of models based on HU features and midline shift for the three classifications of ICP.

	HU Model	Midline Shift Model
All TBI patients (*n* = 47)		
Accuracy	61.70%	40.43%
Precision *	61.06%	32.90%
Recall *	61.70%	40.43%
F1 Score *	0.61	0.29
Unilateral TBI patients (*n* = 34)		
Accuracy	64.71%	41.18%
Precision *	64.02%	16.96%
Recall *	64.71%	41.18%
F1 Score *	0.64	0.24

* The precision, recall, and F1 score were calculated using weighted averages of each category.

## Data Availability

All data generated during this study are included in this article.

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
