# Peer review of "Evaluation of Intracranial Hypertension in Traumatic Brain Injury Patient: A Noninvasive Approach Based on Cranial Computed Tomography Features"

_jcm, 2021, doi:10.3390/jcm10112524_

Round 1

Reviewer 1 Report

The article is very interesting to the clinical practice and gave interesting results. However it needs some improvements : 

1) In the discussion you should cite some cases comparing two or three methods evaluated in the study for the lectures have an objective ideas about the advantages of HU method. You need to improve the discussion.

2) In the introduction you should present with one sentence about HU method such as the main technics applied in this method.

3) I encoraje the authors to describe better this method in Material and methods section objectively: I really couldn't understand the main cerebral structures studied by HU and who developed this method.

4) There are some mistakes in the summary: I would exclude the sentence in the lines 19,20 as I stressed in the text because It's not clear to the reader in this section.

5) In the sentence stressed in the lines 45,46 and 47 there are some study results that shouldn't be in the introduction. You should exclude this sentence and insert a small presentation about HU method.

6) There are some problems with abbreviations in the text: please revise it.

7) How you did you chose the limits to the patients age?

8) The kind of ICP monitoring was not described (intra parenchymal, intra ventricular, subdural etc). You should describe it in the material and methods.

9) Lines 105 to 107 should stay in statistical section.

10) Figure 2: there are three methods for four results. Please correct it.

Author Response

Response to Reviewer 1 Comments

Point 1: In the discussion you should cite some cases comparing two or three methods evaluated in the study for the lectures have an objective idea about the advantages of HU method. You need to improve the discussion.

Response 1: Thank you for the suggestion. We now cited literatures about the advantages of HU in clinical usage and adjusted the Discussion section. (lines 230-243)

Point 2: In the introduction you should present with one sentence about HU method such as the main technics applied in this method.

Response 2: Thank you for the suggestion and we have made adjustments to the Introduction section accordingly. HU method was to establish a model by extracting the HU value and the related features derived from it, which can reflect the symmetry, uniformity and local intensity distribution changes of CT images, so as to evaluate intracranial pressure. (lines 49-53)

Point 3: I encoraje the authors to describe better this method in Material and methods section objectively: I really couldn't understand the main cerebral structures studied by HU and who developed this method.

Response 3: Thank you for the concern. The HU method has been explained in detail in the Discussion section. (lines 230-243)

Point 4: There are some mistakes in the summary: I would exclude the sentence in the lines 19,20 as I stressed in the text because It's not clear to the reader in this section.

Response 4: Thank you for the concern. The regularized results (lines 19-20) have been excluded from the summary according to your suggestion. Considering the relatively small sample size, L2 regularization was used to regularize the logistic regression model of HU-related features. The regularized model added penalized pattern to the HU model, which can reduce overfitting and increase the reliability of the results.

Point 5: In the sentence stressed in the lines 45,46 and 47 there are some study results that shouldn't be in the introduction. You should exclude this sentence and insert a small presentation about HU method.

Response 5: Thank you for the suggestion. We have revised the Introduction. The presentation about HU method has been inserted. (lines 45-53)

Point 6: There are some problems with abbreviations in the text: please revise it.

Response 6: Thank you for the suggestion. We have revised the abbreviations. (lines 30-50)

Point 7: How did you choose the limits to the patients age?

Response 7: Thank you for your concern. In this study, people aged between 18 and 65 were selected for the study, in order to exclude the bias caused by the immature brain tissue in the minors and the age-related brain changes in the elderly. (lines 311-313)

Point 8: The kind of ICP monitoring was not described (intra parenchymal, intra ventricular, subdural etc). You should describe it in the material and methods.

Response 8: Thank you for your concern. We all used the intraventricular sensor, which is now clarified it in the Methods part. (line 80)

Point 9: Lines 105 to 107 should stay in statistical section.

Response 9: Thank you for pointing it out, and we have moved this part to the statistical section. (lines) The description of the evaluation methods for each model has been written in the statistical analysis. (lines 111-114, lines 133-136)

Point 10: Figure 2: there are three methods for four results. Please correct it.

Response 10: Thank you for your concern. In our study, two curves would be generated before and after the regularization of HU Model. These two curves essentially belonged to the HU model, so three methods yielded 4 curves (HU model before regularization, HU model after regularization, midline shift model and clinical expertise model).

Reviewer 2 Report

The review of article jcm-1186563 under the title: Evaluation of intracranial hypertension in traumatic brain injury patients: a non0invasive approach based on cranial computed tomography features.     

It is an interesting and well-written article, and the English should be corrected (nevertheless, I cannot be considered as English native). In this study, Shan and colleagues documented the relationships between Hounsfield unit and the level of intracranial pressure in patients treated for traumatic brain injury. Despite the promising findings, I have one comment.

The authors explain the HU abbreviation only in the abstract. It must also be explained in the main text.

Lines 44-47. The sentence “this report study recruited 47 patients…. “ should be transferred to the result section. This is the result of the authors’ study and not an introduction. The number of studied patients should be reported in the Results section, not in the material and methods. When the authors plan the clinical study they don’t know how many patients (sex, age) they will be enrolled in the study. On the other hand, when the authors plan the experimental observation, they should determine the number of animals.

I think the authors implement intracerebral sensor to measure ICP – not “probe”

The Material and methods section should be rewritten. The authors mix results with methods. The authors cannot speculate in the material and methods section that they include a small sample in their study. (line 97).

Age of participants cannot be presented as a mean and SD. A year cannot be divided into 10. A year has 12 months. I suggest presenting the age as a median with quartile 1 and 3 or as minimum and maximum.

The number of patients should be deleted from the beginning of the discussion. The authors repeat this information 3 times.

The discussion is rather poor. Why the authors did not compare their study with the study of Inaba and colleagues (Am Surg 2007). It was a similar study. Similarly, the authors should compare CT with HU and others methods commonly used for the diagnosis of intracranial hypertension.

There are a lot of mistakes in the citation of articles. For example Bonds and colleagues did not analyze a poor outcome of elevated ICP but they analyze a relationship between hyperthermia after TBI, intracranial hypertension, and poor outcome. Carney and colleagues did not analyze a relationship between intracranial hypertension and poor outcome. They p[resented the guidelines for the management of severe TBI. Reference 15. The name of the author is EIDE. There are a lot of mistakes in the list of authors in the cited references. For example reference 8 – there is a lack of a second author. Additionally, it is a narrative review of non-invasive methods for the measurement of intracranial hypertension in TBI patients. The information in this article does not correspond to the citation.   

Punctuation should be corrected in all text.

References. The name of all cited journals are missing.

Author Response

Response to Reviewer 2 Comments

Point 1: The authors explain the HU abbreviation only in the abstract. It must also be explained in the main text.

Response 1: Thank you for pointing it out. The HU abbreviation is now explained in the Introduction section (Line 50) where it first appeared in the main text.

Point 2: Lines 44-47. The sentence “this report study recruited 47 patients…. “should be transferred to the result section. This is the result of the authors’ study and not an introduction.

Response 2: Thank you for the suggestion. The sentence has been transferred to the Result section according to your suggestion.

Point 3: The number of studied patients should be reported in the Results section, not in the material and methods. When the authors plan the clinical study, they don’t know how many patients (sex, age) they will be enrolled in the study. On the other hand, when the authors plan the experimental observation, they should determine the number of animals.

Response 3: Thank you for the suggestion. We have revised the Methods section.

Point 4: I think the authors implement intracerebral sensor to measure ICP – not “probe”

Response 4: Thank you for the suggestion. We have changed "probe" to "sensor". (lines 77-81)

Point 5: The Material and methods section should be rewritten. The authors mix results with methods. The authors cannot speculate in the material and methods section that they include a small sample in their study.

Response 5: Thank you for the suggestions. We have rewritten the Material and methods section, and the numbers of patients in this part were deleted.

Point 6: Age of participants cannot be presented as a mean and SD. A year cannot be divided into 10. A year has 12 months. I suggest presenting the age as a median with quartile 1 and 3 or as minimum and maximum.

Response 6: Thank you for the suggestion. The age is now presented in median and quartile. (lines 140-141)

Point 7: The number of patients should be deleted from the beginning of the discussion. The authors repeat this information 3 times.

Response 7: Thank you for your suggestion. We have deleted the number of studied patients and have revised the Discussion section.

Point 8: The discussion is rather poor. Why the authors did not compare their study with the study of Inaba and colleagues (Am Surg 2007). It was a similar study. Similarly, the authors should compare CT with HU and others methods commonly used for the diagnosis of intracranial hypertension.

Response 8: Thank you for pointing it out. The Discussion has been rewritten. We have compared HU method with others methods commonly used for the diagnosis of intracranial hypertension in the discussion. The Inaba’s study you mentioned has been added to the citation. (lines 230-243)

Point 9: There are a lot of mistakes in the citation of articles. For example, Bonds and colleagues did not analyze a poor outcome of elevated ICP but they analyze a relationship between hyperthermia after TBI, intracranial hypertension, and poor outcome. Carney and colleagues did not analyze a relationship between intracranial hypertension and poor outcome. They presented the guidelines for the management of severe TBI. Reference 15. The name of the author is EIDE. There are a lot of mistakes in the list of authors in the cited references. For example, reference 8 – there is a lack of a second author. Additionally, it is a narrative review of non-invasive methods for the measurement of intracranial hypertension in TBI patients. The information in this article does not correspond to the citation.

Response 9: Thank you for your concern and we now double checked all the references. The two incorrectly cited papers (line 251,253) and the review literature (line 44) have been replaced. The list of authors in the cited references has been adjusted.

Point 10: Punctuation should be corrected in all text.

Response 10: Thank you for your concern and we now double checked all the punctuation in the text.

Point 11: References. The names of all cited journals are missing.

Response 11: The name of all cited journals has been added.

Reviewer 3 Report

The article deals with a topic of interest although recurrent in the medical literature, such as the search for a non-invasive method, reliable and accurate, for intracranial hypertension prediction.  So far, in the management of traumatic brain injury (TBI), none of the methods has succeeded in replacing invasive intracranial pressure (ICP) monitoring as a standard either to ensure the presence of intracranial hypertension or to guide its treatment. The applicability of this method is exclusively reduced to indicating patients with probability of having intracranial hypertension or at risk of developing it. In any case, for this purpose, simpler, repeatable, inexpensive and less invasive bedside methods like TCD are available. As the authors might know from their experience, TBI is a heterogeneous and dynamic entity; many patients without initial intracranial hypertension develop it in the course of the process, mainly due to new secondary lesions of intracranial or systemic origin. Likewise, patients with extra-axial lesions, once evacuated the lesions, may show normal ICP values. Regardless of these considerations, the article requires a series of modifications and clarifications.

In the introduction, for instance, the authors should not anticipate the results of the research. On the contrary, they should state that they are comparing three non-invasive methods to estimate ICP. In this same section, it is unnecessary to display data that is repeated in the results section, such as the number of patients included, etc. Instead, the introduction must clearly state the purpose of the study.

Among other limitations of design, the study counts a very short series of patients, retrospectively and, in a single center, where patients older than 65 years have not been included. The latter implies that the results cannot be applied to the population as a whole. On the other hand, it is not clarified in the text if the ICP was recorded, in focal lesions, before or after surgery. This information is very important, because in most cases of extra-axial lesions such as epidural hematoma, in most cases, once the hematoma has been removed there is no intracranial hypertension. In the “Results” section, reference is made to the validity of the results in 63 TBI patients with unilateral lesions. They are not described in "Study design and setting”. In my opinion, classification into three groups: <22 mmHg; 23-29 mmHg and> 30 mmHg to evaluate the method is not the most appropriate therapeutic guide. With 23 mmHg of ICP the indication to start active treatment may be delayed while if it is greater than 25 mmHg, the indication of specific treatment is mandatory.

Both in the discussion and in the conclusions, it would be more precise to point out that the proposed method it is possible to evaluate, at a given moment, the ICP values ​​in a semi-quantitative way, given the fact the ICP figures are quantified in ordinal values.

Author Response

Response to Reviewer 3 Comments

Point 1: In the introduction, for instance, the authors should not anticipate the results of the research. On the contrary, they should state that they are comparing three non-invasive methods to estimate ICP.

Response 1: Thank you for your suggestion. We have rewritten this part and made adjustments accordingly. (lines 53-57)

Point 2: In this same section, it is unnecessary to display data that is repeated in the results section, such as the number of patients included, etc. Instead, the introduction must clearly state the purpose of the study.

Response 2: Thank you for your suggestion. We have deleted the number of studied patients except in the result, and the purpose of the study is stated in the introduction. (lines 45-53)

Point 3: Among other limitations of design, the study counts a very short series of patients, retrospectively and, in a single center, where patients older than 65 years have not been included. The latter implies that the results cannot be applied to the population as a whole.

Response 3: Thank you for your concern. This study only included a small sample of single-center adult patients, which were our limitations stated in Discussion section. Further studies were required to assess whether the current result is applicable in the wider range of patients. (lines 303-305, lines 311-314)

Point 4: On the other hand, it is not clarified in the text if the ICP was recorded, in focal lesions, before or after surgery. This information is very important, because in most cases of extra-axial lesions such as epidural hematoma, in most cases, once the hematoma has been removed there is no intracranial hypertension.

Response 4: Thank you for this critical concern. The ICP value was recorded during surgery before craniotomy or decompressive craniectomy, which was clarified in the Material and method section as “The initial ICP was recorded during operation immediately after the insertion of ICP sensor (Integra, USA).” (line 77)

Point 5: In the “Results” section, reference is made to the validity of the results in 34 TBI patients with unilateral lesions. They are not described in "Study design and setting”.

Response 5: Thank you for your concern. The numbers and overall characteristics of patients enrolled including number of patients with unilateral lesions were presented in the first paragraph of Result section.

Point 6: In my opinion, classification into three groups: <22 mmHg; 23-29 mmHg and> 30 mmHg to evaluate the method is not the most appropriate therapeutic guide. With 23 mmHg of ICP the indication to start active treatment may be delayed while if it is greater than 25 mmHg, the indication of specific treatment is mandatory.

Response 6: Thank you for pointing it out. The threshold of 22 mmHg was made according to E. Sorrentino and colleagues’ study on critical thresholds for cerebrovascular reactivity after TBI, which was associated with increased mortality rather than indication for operation. (lines 250-251)

Point 7: Both in the discussion and in the conclusions, it would be more precise to point out that the proposed method it is possible to evaluate, at a given moment, the ICP values in a semi-quantitative way, given the fact the ICP figures are quantified in ordinal values.

Response 7: We have made the revisions in the discussion and conclusions based on your suggestion. (line 310,319)

Round 2

Reviewer 1 Report

I think that changes in the research are enough.

Author Response

Thank you for your approval.

Reviewer 2 Report

I have no comments

Author Response

Thank you for your approval.

Reviewer 3 Report

Thanks for your attention to the comments  and suggestions made. I believe, that some aspects that could lead confusion among readers have been clarified. However, in the conclusions, it would be appropiate to point out, as have been mentioned in the discussion section, that the most suitable application of this method, is to predict which patients are at risk of suffering intracranial hypertension, I encourage the authors to continue investigating, with a larger population sample,  the best applications of the proposed method.
